# The Effect of Static Apnea Diving Training on the Physiological Parameters of People with a Sports Orientation and Sedentary Participants: A Pilot Study

**DOI:** 10.3390/sports12060140

**Published:** 2024-05-22

**Authors:** Dmitriy Bezruk, Petr Bahenský, David Marko, Miroslav Krajcigr, Petr Bahenský, Eva Novák-Nowická, Tomáš Mrkvička

**Affiliations:** 1Department of Sports Studies, Faculty of Education, University of South Bohemia in České Budějovice, 37005 České Budějovice, Czech Republic; bezrud00@jcu.cz (D.B.); dmarko@pf.jcu.cz (D.M.); mkrajcigr@pf.jcu.cz (M.K.); petrbahensky@email.cz (P.B.J.); enowicka@pf.jcu.cz (E.N.-N.); 2Department of Applied Mathematics and Informatics, Faculty of Economics, University of South Bohemia in České Budějovice, 37005 České Budějovice, Czech Republic; mrkvicka.toma@gmail.com

**Keywords:** breathing, breath holding, forced vital capacity, heart rate, oxygen saturation, runners, sedentary participants, swimmers

## Abstract

Diver training improves physical and mental fitness, which can also benefit other sports. This study investigates the effect of eight weeks of static apnea training on maximum apnea time, and on the physiological parameters of runners, swimmers, and sedentary participants, such as forced vital capacity (FVC), minimum heart rate (HR), and oxygen saturation (SpO_2_). The study followed 19 participants, including five runners, swimmers, sedentary participants, and four competitive divers for reference values. The minimum value of SpO_2_, HR, maximum duration of apnea, and FVC were measured. Apnea training occurred four times weekly, consisting of six apneas with 60 s breathing pauses. Apnea duration was gradually increased by 30 s. The measurement started with a 30 s apnea and ended with maximal apnea. There was a change in SpO_2_ decreased by 6.8%, maximum apnea length increased by 15.8%, HR decreased by 9.1%, and FVC increased by 12.4% for the groups (*p* < 0.05). There were intra-groups changes, but no significant inter-groups difference was observed. Eight weeks of apnea training improved the maximum duration of apnea, FVC values and reduced the minimum values of SpO_2_ and HR in all groups. No differences were noted between groups after training. This training may benefit cardiorespiratory parameters in the population.

## 1. Introduction

Breathing is one of the primary automatic processes in the body. It is crucial not only for supplying oxygen and removing carbon dioxide from the body but also mainly affects the pH balance of the blood and supports the proper functioning of vital organs. Breathing is linked to many other biological processes, such as regulating the heart rate (HR) and the nervous system, which points to its complexity and importance in maintaining the vitality of the individual [1,2]. Nowadays, more and more experts are focused on studying breathing parameters, patterns, and strategies that can influence these patterns. This effort to understand breathing leads to the development of new therapeutic approaches and techniques that can be used to promote health and treat various conditions, including anxiety, stress, and sleep disorders [3,4,5,6]. This topic is emphasized due to the increasing standard of living and changes in the population’s movement habits. These changes can affect the involvement of respiratory muscles and, to some extent, postural muscles, which has the potential for prolonged health consequences [7].

Regular breathing exercises have a positive effect on increasing endurance and improving overall mental and physical health in athletes and the entire population. These exercises contribute to increasing the efficiency of breathing, which positively impacts the cardiovascular system and overall vitality [8]. The quality of breathing plays a decisive role in performance in many sports disciplines. The direct effect on performance is particularly evident in the field of endurance sports, where the ability to maintain an optimal breathing rhythm can significantly improve results. However, in other sports, the breathing process is of fundamental importance, whether it is movement coordination, endurance, or concentration. Top athletes often focus on improving performance and achieving goals through systematic breathing techniques and exercise training [9]. The effect of hypoxia on respiratory parameters in sports practice is well documented, for example, a decrease in SpO_2_ values or an increase in minute respiratory volume [10]. Scientific studies have examined this phenomenon, which has practical applications in athletes’ training programs. For example, this effect is also observable during training at altitudes, where reduced oxygen availability affects breathing patterns and respiratory function. Similar effects can also be simulated at home using a hypoxia simulation device. In addition, athletes can engage in hypoventilation or apnea training, which are other methods that can positively affect their breathing abilities and overall physiology [11]. The practice of breath holding, also referred to as hypoventilation, CO_2_ tolerance, or air hunger training, has gained significant attention recently, attributed to substantial performance enhancements reported in swimming and methodologies promoted by freediving [12]. Apnea exercises of varying intensity and duration are common practice in athletes’ training programs and are considered a standard part of their training. The apnea duration usually varies depending on whether the condition is static or dynamic. In static apnea, which is typical for divers, the duration is longer than in dynamic conditions. In this discipline, known as Static Apnea, we have experienced an unprecedented increase in popularity over the last three decades. Static Apnea has become an internationally recognized sport, gaining increasing interest from athletes worldwide. Thanks to ever-developing techniques and training methods, new records and limits are being set in this exciting discipline [13,14,15]. During this time, the static apnea record has improved, doubling since it was first recorded. The current static apnea record is 11 min 35 s [16]. Apnea has become a popular discipline in various sports, from traditional sports to the latest ones, such as fish photography. This diversity demonstrates the multifaceted uses of this skill and discipline [17,18,19,20].

Apnea is often considered a model of reduced oxygen availability that can be spontaneously compared to hypoxia as a model of a hypoxic stimulus. This provides valuable insights for investigating the body’s physiological responses to oxygen deprivation. Studying the body’s reactions to apnea and hypoxia allows for a deeper understanding of the adaptive mechanisms involved in changes in the respiratory environment. These findings have wide application in medicine, scientific research, and sports training, where strategies are sought to optimize performance and health in conditions of reduced oxygen [21,22]. During voluntary apnea, there is a reaction aimed at maintaining critical physiological functions in the body. These reactions rely on oxygen stored in the lungs, blood, and other tissues to maintain the oxygen supply for essential life processes [23,24]. The diving response is a complex set of physiological adaptations that occur due to immersion in water. These adaptations include bradycardia, peripheral vasoconstriction, increased arterial blood pressure, decreased cardiac output and blood flow, and increased sympathetic nervous system activity. The cessation of ventilation induces these changes and is crucial to optimizing the diver’s physiological processes in conditions of low oxygen and increased water pressure [25,26,27,28]. For divers who can consciously suppress the urge to breathe, achieving the most extended possible breath hold while maintaining consciousness rests on two primary principles. The first principle is an increased oxygen transfer capacity, often related to an increased lung volume, thereby allowing a more excellent oxygen supply to the body. The second principle is the maintenance of available oxygen in the organism, which is achieved thanks to the central distribution of blood flow and the reduced rate of oxidation of metabolic processes [17]. The diving response, often assessed using a bradycardia scale, indirectly indicates a decrease in arterial oxygen saturation. This means that the more significant the reduction in blood oxygen saturation during diving, the more bradycardia increases as the body’s response compensates for insufficient oxygen [17,29,30,31]. Apnea training, especially breath holding (BH), aims to increase the maximum apnea time. However, endurance training is essential to achieve high performance in this sport. This includes physical training with a combination of strength and cardiorespiratory exercises and a combined form of exercise combining BH with physical training. It is important to note that there is no single ideal method for improving static apnea time, so it is often advisable to experiment with different training approaches and techniques to achieve optimal results [18]. Also, static apnea training is divided into underwater or dry training [18]. Static apnea training has the potential to effectively reduce the body’s response to stressful situations, which can subsequently help reduce the body’s stress response in challenging situations. This type of training also provides valuable benefits in heart rate regulation during apnea, leading to better breath control and overall higher diving performance [32]. Static apnea training is known to help improve breath control and overall diving performance. In addition, such training has been shown to increase blood hematocrit levels, indicating an improved ability of the body to transport oxygen and other nutrients to cellular tissues [33]. This phenomenon has important implications, especially in swimming, where efficient oxygen transport to the tissues is crucial for optimal performance. Therefore, performing apnea training just before swimming can positively affect a swimmer’s performance [34,35]. Furthermore, splenic contractions may have a role in rapid adaptation to altitude or other ambient hypoxic conditions, serving as a compensatory mechanism by bridging the gap between non-acclimation and altitude-induced polycythemia [33,36].

Interindividual differences in HR during apnea can show a wide range of variability, from negligible to a decrease in HR up to 70–80% of the pre-apnea HR value. Some divers may experience a significant reduction in HR to values between 20 and 30 beats per minute [32,37,38,39]. Unfortunately, the most critical factor for divers is their maximum apnea time per breath [18,40,41,42,43,44,45]. Our study aimed to determine how eight weeks of static apnea training used by divers affects physiological cardiorespiratory parameters in athletes and individuals who do not participate in regular physical activity. We assume that the eight-week static apnea training will positively affect the physiological parameters of all groups. Specifically, it will reduce SpO_2_ and HR values, increase the maximum duration of apnea, and increase FVC values.

## 2. Materials and Methods

### 2.1. Participants

In total, the study involved 20 participants (males) of various specialties, of which 15 participated in the intervention. During the measurement, one diver became ill for a long time and could not participate in the study. The intervention groups of participants never completed any static apnea training. The participants consisted of three intervention groups of participants as follows: competitive middle- and long-distance runners (RUN; *n* = 5; 20.8 ± 5.7 years; 74.6 ± 10.5 kg; 189.4 ± 4.6 cm), competitive swimmers (SWIM; *n* = 5; 18 ± 1.0 years; 75.1 ± 11.1 kg; 181.4 ± 7.1 cm), five sedentary participants who were not active members of any sports club in the last five years and were not regularly physically active (SED; *n* = 5; 18.2 ± 1.6 years; 71.4 ± 15.2 kg; 178.2 ± 8.5 cm). To compare the results, as reference values, the results of professional divers who have been doing the selected training for at least five years (DIV; *n* = 4; 26.5 ± 3.1 years; 83.9 ± 13.0 kg; 182.5 ± 3.1 cm) were used. All participants completed a written informed consent. There was no compensation for any of the participants, and all protocols and procedures conformed to the Declaration of Helsinki statements and were approved by The Ethical Committees of the Faculty of Education, University of South Bohemia study on 19 October 2018 (002/2018). The criteria for including participants in the research were as follows: sports participants had to train regularly, at least five times a week, for three years. All participants had to be non-smokers and in good health. They were also asked if they would complete eight weeks of static apnea exercises at home. To ensure that participants were not at increased risk of health complications, there was no record of respiratory illness either three weeks before the start of the study or during its follow-up.

### 2.2. Experimental Design

During the measurement, participants completed initial laboratory tests, an eight-week home intervention, and final laboratory tests after the intervention was completed. These tests and interventions consisted of static apnea diving training on land. The eight-week intervention took place in the home environment and was identical to the initial and final measurements in the laboratory. The participants performed breathing exercises four times a week, specifically on Monday, Tuesday, Thursday, and Friday (see subsection Breathing exercises). After each training unit, each participant was sent information about the completed training and the duration of maximum apnea. In the laboratory test, forced vital capacity (FVC) was measured using by breath-by-breath metabolic analysis (Metalyzer B3, Cortex, Leipzig, Germany), after which the same breathing exercises as during training were performed on the couch. When completing the test at the end of the maximum apnea and during the test, the following parameters were measured: saturation (SpO_2_) using the Prince–100 B Fingertape Oximeter (Heal Force, Shanghai, China), HR using the Polar chest strap (Polar, Guangzhou, China), and the maximum duration of apnea that was measured by the laboratory technician with a stopwatch. This study is focused on results at the end of maximal apnea only. Only an initial examination was conducted for the group of divers because they carried out the training for a long time.

### 2.3. Breathing Exercise

Measurements before and after the intervention and the intervention itself were the same. The test occurred in the laboratory lying on a couch at a room temperature of 20.5–21.5 °C. The participants were not wearing any masks or any respiratory protection. During the measurement, we used apnea diving training based on O_2_ training [46], which was modified by the divers who participated in the study over many years of training. All subject groups began with 30 s of apnea followed by 60 s of hyperventilation. Hyperventilation intervals were according to the ambient air (FiO_2_ = 20.93%). The participants’ hyperventilation time between apneas was always the same at 60 s. Apnea duration between hyperventilations was consistently increased by 30 s, and the sixth terminal apnea was maximal for all participants; the measurement scheme is described in Figure 1.

### 2.4. Statistical Analysis

MANOVA was used to compare individual groups of athletes in all indicators. Specifically, its implementation using global envelopes [47], which allows us to both globally compare individual groups of athletes in all indicators at the same time, and graphically determine which indicators are responsible for a positive test result. This graphical interpretation is made possible by envelopes (non-parametric method) that determine the non-rejection region at the 0.05 significance level. Thus, an indicator outside these envelopes is demonstrably significant at the global significance level of 0.05. In addition to comparing individual groups before and after training, we also compared the differences in results before and after training. Although these differences are non-significant, the graphical interpretation shows trends that we expect would be statistically significant if more participants were available. Between sessions comparison of parameters (separately intra-group and for all participants together) were also performed using a Wilcoxon pair test. Statistical analysis was conducted using Excel 2016 (Microsoft Corporation, Redmond, WA, USA) and IBM SPSS Statistics (IBM, Armonk, NY, USA) for Windows platform. Through Sensitivity Power Analysis, we found that the model (Power = 0.8; α = 0.05) used in the pilot study reveals differences in effect size f ≥ 0.456.

## 3. Results

Table 1 summarizes pre- and post-intervention results for each group separately and pooled results for all groups combined. This table also includes the results of professional divers, allowing comparison of results between study groups and professional divers. As a result of the intervention program, the minimum value of SpO_2_ decreased in all groups of participants. This change was significant (*p* < 0.05) only for the swimmers and sedentary participants, not the runner group. The intervention program significantly affected (*p* < 0.05) by reducing HR values in swimmers, prolonging maximum apnea time in all groups of participants, and FVC values in swimmers and sedentary participants. The effect of diving training was significant (*p* < 0.05) for SpO_2_, HR, FVC, and maximum apnea duration for all groups combined. The results of the professional divers compared to the other groups were significantly higher. However, the difference between the intervention groups and the divers decreased in all groups due to the intervention program.

Figure 2 present and evaluate the differences in observed values between groups. Figure 2a shows the results of the measurements before the intervention and the difference between the groups. The red point indicates the significance of the difference compared to the other groups. When comparing Figure 2a,b, it is clear that the differences between the groups have decreased due to the completed training. The results of the minimum SpO_2_ and the maximum apnea duration in professional divers were significantly higher before the intervention compared to the other groups. On the contrary, the group of sedentary participants achieved significantly lower FVC values before the intervention. Figure 2c shows the intervention’s effect and compares the effect between the intervention groups. In the sedentary participants, SpO_2_, FVC, and maximum apnea duration improved more after the intervention than in other runners and swimmers. The swimmer group had the most significant improvement in HR scores. The global significance of the intervention is *p* = 0.085, which is very promising for the given number of participants in each group.

## 4. Discussion

Our study examined the effect of eight weeks of diving training on land in groups of runners, swimmers, and sedentary participants. Furthermore, their results were compared with those of professional divers, which were also measured in the laboratory and used in the given study only for comparison. Although this study is a pilot, its results can be useful for more extensive research. Swimmers achieved the lowest saturation during the initial testing compared to the other groups of intervention participants, but the difference was not significant. When comparing HR input values, runners achieve significantly the weakest values, even comparable to divers. Swimmers achieve the highest values; we have no explanation for this fact. The length of entry maximum apnea is the highest in swimmers and the lowest in sedentary participants. Swimmers and runners achieve higher FVC values in the entry test than sedentary participants. The pre-intervention values of FVC of all intervention participants were significantly lower than those of professional divers; the most marked difference was noted in the duration of apnea and minimum SpO_2_ values. As a result of the intervention, this difference was reduced, regardless of the group affiliation of the participants. Significant changes in physiological parameters occurred when investigating the effect of eight weeks of static apnea on different focus groups of participants. As a result of the breathing training, there was an increase in the maximum duration of apnea (15.8%), a significant decrease in SpO_2_ value (6.8%), an increase in FVC values (12.4%) and decrease in HR values (9.1%) (*p* < 0.05).

Furthermore, we presented the differences in selected cardiorespiratory parameters between the exercising and sedentary populations and the possibility of influencing them. Consistent with previous findings, this study suggests improvements in apnea duration and SpO2 reductions in endurance and sedentary participants previously noted in divers [28]. Traits of elite free-divers indicate that prolonged adjustments to breath-holding entail diminished sensitivity to CO_2_ and amplified lung capacity [17,48]. It is suggested that untrained individuals and endurance athletes can significantly affect SpO_2_ troughs, HR, apnea duration, and FVC values through targeted apnea training. Hyperventilation increases the drop in SpO_2_ more than average ventilation [49]. A decrease in SpO_2_ was recorded during apnea, which increased depending on the apnea duration [22]. We also know that the decline in SpO_2_ changes with the level of apnea experience, with more experienced people experiencing a more pronounced decrease [50,51]. Similarly, divers who can hold apnea for more than 4 min have a modest increase in ischemic-modified albumin and more excellent resistance to hypoxia compared with divers who can hold apnea for less than 4 min [50,52]. This study suggests that a decrease in SpO_2_ is also possible during the maximum duration of static apnea. This training could probably lead to a more pronounced desaturation of the organism, which could subsequently indicate a better resistance to hypoxia. This would be consistent with the literature that reports that regular apnea training confers significant cardiorespiratory benefits and increases tolerance to hypoxia and hypercapnia, thereby providing participants with better resistance to breath-hold stress [53].

During apnea, HR decreases due to increased activity of the parasympathetic nervous system, which dampens the heart, and concomitant vasoconstriction occurs, which decreases HR [54,55]. We can hypothesize that with dry conditions, apnea-associated bradycardia may have a more pronounced effect than exercise-associated tachycardia [56]. Although the change in HR in all groups was significant overall in our study, in the individual comparison of the groups, only the swimmers achieved a substantial decrease in HR. It can be assumed that the swimmers were more relaxed during apnea than the other groups because they are used to working with breath holding and the mental side of their practice. Mental stress increases HR and, thus, oxygen consumption [14]. Another possible explanation for why the swimmers’ heart rates dropped so much is that after consulting with the coach, we found three of the five swimmers had tachycardia. Breathing exercises positively affect respiratory functions and strengthen the respiratory muscles [57].

Increasing levels of CO_2_ in the body may have a bronchodilatory effect and potentially alleviate bronchoconstriction [58]. In our study, FVC improved in all participating groups. The results suggest that an increase in lung volume is one of the respiratory adaptations observed in trained divers after apnea and is attributed to respiratory muscle training resulting from more intensive breathing during diving [54,57]. Another positive result was an increase in the maximum time participants could hold their breath. This phenomenon was observed in all group participants after the intervention and can be attributed to the strengthening of respiratory muscles, adaptation to apnea training, and increased tolerance to hypoxia [50,54]. Static apnea training improved all intervention group participants, and no significant differences between groups were noted after the intervention. This fact could indicate the effectiveness of training for sport-oriented participants, who may gain additional benefits in their discipline, and for the general population, which could indicate the potential of training interventions to improve individuals’ overall physical fitness and health across different activities and lifestyles. Training can help both swimmers, where static apnea reduces the drop in SpO_2_ during swimming [59] and the general population. The results of general and specific groups differ significantly from those of professional divers who have been training for breathing for several years. Their results are considerably better in all parameters. This study had some limitations such as the time-consuming nature of the laboratory test, the small number of participants, and the lack of female participation. This may affect the generalizability of the obtained results.

However, this pilot study provides valuable initial insights and makes it possible to identify potential directions for further research. Further research should include a more diverse and more extensive group of participants. Also, a more extended intervention could bring more significant changes as breath-holding training accumulates over time [28]. Furthermore, it can be assumed that potential sex differences in the effects of apnea training could reveal new aspects of physiological adaptation to training. Also, tailoring training interventions to specific sports or populations (e.g., asthmatics) can maximize the potential of apnea training. It is also recommended that a performance test be added to further research. Based on the results of this study and best practices, a more extensive and comprehensive analysis will follow, allowing a deeper understanding of the investigated issue.

## 5. Conclusions

An eight-week static apnea-based breathing exercise intervention resulted in remarkable changes in maximal apnea duration and significant reductions in HR, decreases in SpO_2_ values, and increases in FVC values. These effects were observed in high-performance swimmers, runners, and sedentary populations, suggesting universal benefits of this type of exercise for different groups of individuals with varying physical activity levels. There is no significant difference in the level of changes in individual groups. We have confirmed that the training used by competitive divers using breath holding on land has a significant effect on selected cardiorespiratory parameters of the sports and sedentary population. This intervention could improve respiratory parameters in the general population and athletes, where oxygen intake and carbon dioxide tolerance are directly related to performance.

## Figures and Tables

**Figure 1 sports-12-00140-f001:**
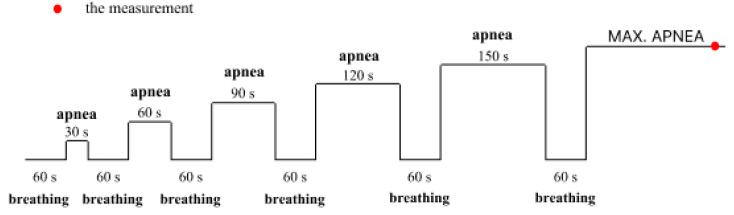
Illustration of static apnea diving training and the course of measurements in the laboratory.

**Figure 2 sports-12-00140-f002:**
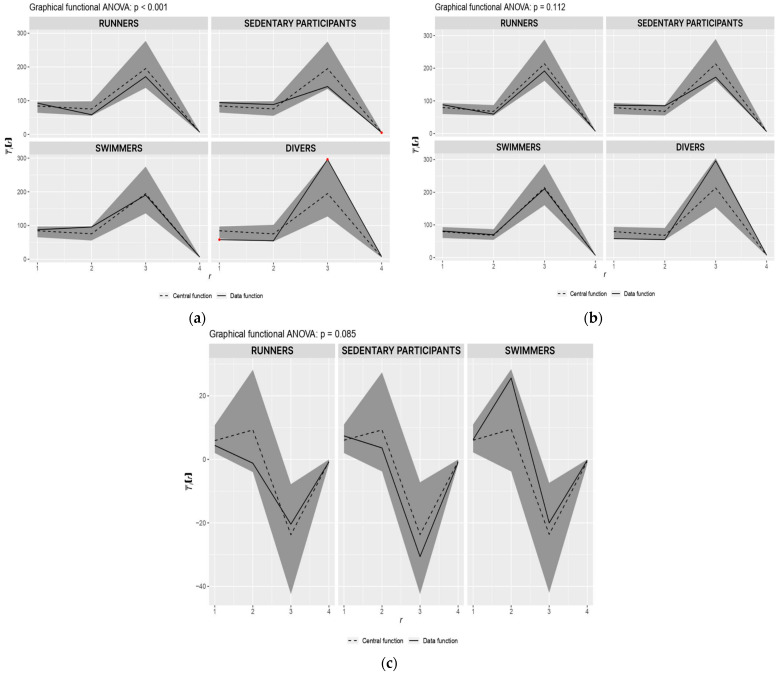
Comparison of the significance of the result of all groups of participants and global significance of intervention: (**a**) Comparison of the significance of the results of all groups of participants before the intervention; (**b**) Comparison of the significance of the results of all groups of participants after the intervention; (**c**) Comparison of the effectiveness of diving training for groups of runners, swimmers, and sedentary participants. Note: On the x–axis, 1 is SpO_2_, 2 is HR, 3 is the duration of maximum apnea, and 4 is FVC.

**Table 1 sports-12-00140-t001:** Results of all groups of participants at the end of maximum apnea before and after the intervention and comparison with the group of divers.

Parameters	Group	Before Intervention	After Intervention	Mean Difference	% Change	% Change All Participants
SpO_2_ (%)	RUN	92.6 ± 3.6	88.2 ± 5.1	−4.4	4.8	
	SWIM	87.4 ± 10	81.2 ± 10.1	−6.2 *	7.1	6.8 #
	SED	94.2 ± 7.1	86.8 ± 14.5	−7.4 *	8.6	
	DIV	58 ± 13.9				
HR (bpm)	RUN	58.2 ± 5.8	59.4 ± 5.1	1.2	−2.1	
	SWIM	95.6 ± 10.2	70.0 ± 13.0	−25.6 *	26.8	9.1 #
	SED	88.4 ± 15	84.8 ± 16.7	−3.6	4.1	
	DIV	54.8 ± 4.4				
Apnea (s)	RUN	171.2 ± 15.8	191.6 ± 11.1	20.4 *	11.9	
	SWIM	190.4 ± 32.5	210.4 ± 36.5	20.0 *	10.5	15.8 #
	SED	150.0 ± 41.4	172.6 ± 43.4	30.6 *	21.5	
	DIV	296.3 ± 91.8				
FVC (l)	RUN	6.3 ± 0.6	7.1 ± 0.9	0.7	12.2	
	SWIM	6.3 ± 0.5	6.6 ± 0.6	0.2 *	4.7	12.4 #
	SED	5.0 ± 0.3	5.9 ± 0.8	0.9 *	18.4	
	DIV	7.8 ± 1.5				

Note: * *p* < 0.05 for intra-group; # *p* < 0.05 for all participants; SpO_2_: min.saturation; HR: heart rate; FVC: forced vital capacity; RUN: runners; SWIM: swimmers; SED: sedentary participants; DIV: divers.

## Data Availability

Data are contained within the article.

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
