# Peer review of "The Effect of Static Apnea Diving Training on the Physiological Parameters of People with a Sports Orientation and Sedentary Participants: A Pilot Study"

_sports, 2024, doi:10.3390/sports12060140_

Round 1
Reviewer 1 Report
Comments and Suggestions for Authors
The article "The effect of static apnea diving training on the physiological parameters of people with a sports orientation and sedentary participants: a pilot study” aimed to determine how eight weeks of static apnea training used by divers affects physiological cardiorespiratory parameters in athletes and individuals who do not participate in regular physical activity. At the same time, the study compared the results of sporty and sedentary participants with the results of professional divers who used this training long-term and were measured once. The study followed 19 participants, including five runners (n = 5), swimmers (n = 5), sedentary participants (n = 5), and professional divers. The minimum value of oxygen saturation (SpO2), heart rate HR), maximum duration of apnea and vital lung capacity (FVC) were measured. There was change in SpO2 values by 6.8%, maximum apnea length by 15.8%, HR by 9.1%, and FVC by 12.4% for the groups. There were changes in the groups, but no significant difference between individual groups 23 was observed. The authors conclude that eight weeks of apnea training improved the maximum duration of apnea, FVC values and reduced the minimum values of SpO2 and HR in all groups.
The topic of this study is interesting and it provides important findings. However, the study presents a significant methodological limitation regarding the low sample size adopted and, consequently, the statistical procedures used. It is not possible to guarantee that the results are reliable with few subjects in each experimental group. Furthermore, some clarifications and corrections are necessary.
So, for this study to be approved, it needs to be more speculative. Considering it's a pilot study with methodological issues, I do not approve this study (as it is) for publication in MDPI Sports.
Specifical suggestions:
Abstract:
The Abstract is well-written. It is suggested, however, to include the data stratified by groups.
Page 1, line 16: It is suggested VLC.
Page 1, lines 21-23: It is suggested to present the values stratified by groups.
Page 1, line 24: Inter-group differences or intra-group differences was not observed? What is this? (4):
Keywords:
To adopt words that are not in the title and list them in alphabetical order.
Introduction:
The Introduction is interesting and addresses the importance of apnea training. However, it does not cite specific results regarding the improvement of athletic physical capacities. It is suggested to cite specific results from studies, highlighting the physiological mechanisms that result in the enhancement of physical capacities. Additionally, special attention to the text format and the need for new sentences is recommended. Finally, the problem (justification) of the study should be better addressed.
Page 2, lines 54, 64, 72, 77, 84, 95, 99, 104 and 113: A new sentence is suggested.
Page 2, lines 55-57: But are the physiological mechanisms in altitude and apnea the same? And what are the influences of apnea training on athletic physical capacities, such as anaerobic capacity (anaerobic sports)?
Page 3, lines 129-133: The main objective of the study should be only one.
Materials and Methods
Methodologically, it is necessary for the authors to justify the adopted sample size, statistical methods, and scientific basis.
Participants:
Inter-group statistical analyses with low sample numbers can result in Type I errors. What ensures that groups of 4 and 5 samples can represent the populations (middle- and long-distance runners; swimmers; sedentary individuals)? How was the sample size per group determined?
Page 3, line 139: Three or four groups? RUN, SWIM, SED and DIV (although the intervention was for the first three groups mentioned).
Breathing exercise:
Based on which scientific sources was this methodology adopted?
Statistical analysis:
The sample size appears to be very small per group. Was a normality test conducted? In this case, it is believed that it would be best to adopt descriptive statistics only.
Results:
Table 1:
For intra-group statistical differences, adopt (*). For inter-group statistical differences (before and after), adopt (#). Additionally, the table should be self-explanatory, indicating the sample size per group and informing what each acronym corresponds to (RUN, SWIM, SED, and DIV).
Figure 2:
Figure 2 is quite confusing and of poor quality. Also, it is not self-explanatory. It is suggested to omit it from the article. Perhaps adopting scatter plots would be better.
Discussion:
The discussion is quite descriptive and speculative (with little comparison to other scientific studies). Additionally, the discussion may be compromised due to the adopted statistical and sampling procedures. The text is continuous, with few sentences.
Page 7, line 249: professional drives is the control group?
Page 7, lines 251-257: Although it is possible to explain some results, it is believed that these results are compromised due to the low sample size adopted in each group.
Page 7, line 257: A new sentence is suggested.
Conclusions:
The study is interesting but methodological aspects compromise the discussion and conclusion.
Author Response
Response to Reviewer 1 Comments
The authors would like to thank the reviewers for their interest in our work and for their insightful comments. We have carefully considered each of these recommendations and revised our manuscript accordingly using the “track changes” feature, and provided a point-by-point response to each of the individual recommendations below. We are appreciative of the reviewers’ time and energy.
Point 1: Page 1, line 16: It is suggested VLC.
Response 1: Thank you for your comment. The words vital lung capacity have been replaced by forced vital capacity.
Point 2: Page 1, lines 21-23: It is suggested to present the values stratified by groups.
Response 2: Thank you for your comment. In the 1st study assignment, we were advised to shorten the abstract to 200 words, and we had to shorten the abstract to 202 words.
Point 3: Page 1, line 24: Inter-group differences or intra-group differences was not observed? What is this?
Response 3: Thank you for your comment. Text in line 24 was rewritten.
Point 4: Keywords: To adopt words that are not in the title and list them in alphabetical order.
Response 4: Thank you for your comment. We changed the Keywords.
Point 5: Page 2, lines 54, 64, 72, 77, 84, 95, 99, 104 and 113: A new sentence is suggested.
Response 5: Thank you for your comment. All sentences have been changed.
Point 6: Page 2, lines 55-57: But are the physiological mechanisms in altitude and apnea the same? And what are the influences of apnea training on athletic physical capacities, such as anaerobic capacity (anaerobic sports)?
Response 6: The sentence was corrected, and the following sentence had a more detailed explanation of the effect on athletes.
Point 7: Page 3, lines 129-133: The main objective of the study should be only one.
Response 7: Thank you for your comment. The sentence about comparing the results of intervention groups and divers was deleted.
Point 8: Materials and Methods: Methodologically, it is necessary for the authors to justify the adopted sample size, statistical methods, and scientific basis.
Response 8: Thank you for your comment. Everything is explained in the following answers.
Point 9: Participants: Inter-group statistical analyses with low sample numbers can result in Type I errors. What ensures that groups of 4 and 5 samples can represent the populations (middle- and long-distance runners; swimmers; sedentary individuals)? How was the sample size per group determined?
Response 9: Thank you for your comment. We had 5 participants in each group during the measurement, but one diver became ill for a long time and could not participate (it was added to the article). And this study is just a pilot study. Sample size is listed in the article. We used a statistics technique that is based on permutations and thus is valid for a small amount of data.
Point 10: Page 3, line 139: Three or four groups? RUN, SWIM, SED and DIV (although the intervention was for the first three groups mentioned).
Response 10: We added the word intervention to clarify the difference between an intervention group and a diver group.
Point 11: Breathing exercise: Based on which scientific sources was this methodology adopted?
Response 11: Thank you for your comment. We have added the source to the text and corrected it slightly.
Point 12: Statistical analysis: The sample size appears to be very small per group. Was a normality test conducted? In this case, it is believed that it would be best to adopt descriptive statistics only.
Response 12: Data normality does not need to be calculated. Given the sample of participants, we used for statistics non-parametric tests. We know that the sample of participants is small, and therefore, we did not use the standard ANOVA methods, which are susceptible to the amount of data and the normality of the set, and we used a technique that is based on permutations and thus is valid for a small amount of data.
Point 13: Results: Table 1: For intra-group statistical differences, adopt (*). For inter-group statistical differences (before and after), adopt (#). Additionally, the table should be self-explanatory, indicating the sample size per group and informing what each acronym corresponds to (RUN, SWIM, SED, and DIV).
Response 13: Thank you for your comment. Necessary items have been added to the table. The sample size is written in the methodological section.
Point 14: Figure 2: Figure 2 is quite confusing and of poor quality. Also, it is not self-explanatory. It is suggested to omit it from the article. Perhaps adopting scatter plots would be better.
Response 14: Thank you for your comment. The quality of Figure 2 was improved. We believe that this type of graph can suitably express what we need to present.
Point 15: Discussion: The discussion is quite descriptive and speculative (with little comparison to other scientific studies). Additionally, the discussion may be compromised due to the adopted statistical and sampling procedures. The text is continuous, with few sentences.
Response 15: Thank you for your comment. We expanded the discussion and compared the results with those of other studies. Errors in the statistical part have been corrected.
Point 16: Page 7, line 249: professional drives is the control group?
Response 16: In that sentence (line 145–148), it is written that the divers' results were used only for comparison.
Point 17: Page 7, lines 251-257: Although it is possible to explain some results, it is believed that these results are compromised due to the low sample size adopted in each group.
Response 17: Thank you for your comment. It was stated in the limits of the work.
Point 18: Page 7, line 257: A new sentence is suggested.
Response 18: Thank you for your comment. The sentence was rewritten.
Reviewer 2 Report
Comments and Suggestions for Authors
This was a very interesting study. I have a number of suggestions for revision:
Line 16: Indicate what FVC stands for. I think it should be forced vital capacity?
I am not sure the numbering is required in the abstract (i.e. (1), (2), (3), (4))
Line 21: please indicate whether this was an increase or decrease in SPO2 (I assume decrease?)
Line 22: Same comment for HR. Please indicate whether this is an increase or decrease.
Lines 41-42: “The topicality of this topic...” This sounds awkward. I suggest re-wording.
Lines 45-46: “Regular breathing exercises have a positive effect on increasing endurance and improving overall mental and physical health in athletes and the entire population.” References are needed to support this statement.
Lines 48-51: I think references are also needed here to support the statements: “The quality of breathing plays a decisive role in performance in many sports disciplines. The direct effect on performance is particularly evident in the field of endurance sports, where the ability to maintain an optimal breathing rhythm can significantly improve results.”
Line 79: “This insight provides valuable insights for investigating...” I suggest changing this to “This provides valuable insights for investigating...”
Lines 99-101: “The magnitude of the diving response, which is often measured using the bradycardia scale, appears to be inversely proportional to the decrease in arterial oxygen saturation” – should this be “proportional” and not “inversely proportional”?
Lines 105-106: “However, endurance training is essential to achieve high performance in this sport.” – what sport are you referring to here?
Lines 116-117: “...such training has been shown to increase blood hematocrit levels” – please provide a reference for this statement.
Please include a hypothesis statement (or statements) at the end of your introduction.
Line 166: As with the abstract, please spell out / explain the abbreviation FVC.
Line 181: “FiO2 – 20.93%” – should this be “FiO2 = 20.93%”?
When describing your p-values in the manuscript, make sure to use decimal places and not commas (i.e. p<0.05 and not p<0,05).
The statistical analysis section is very unclear. This can be clarified by simply using a 2-factor (group x time, with repeated measures on the time factor) ANOVA with Bonferroni post-hoc testing if you have a group main effect or group x time interaction. It is not clear in the statistics section how the diving group was compared to the other groups. I suggest comparing the post-training results from the other groups to the diving group in a one-factor ANOVA. If this is significant, you can use a Bonferroni test to determine which groups are different.
Lines 211-212: “This change was significant (p<0,05) only for the swimmers and sedentary participants, not the runner group.” You should run a group x time ANOVA. You can only state that there were differences between groups over time if the group x time interaction is significant.
I think overall the results need to be more clearly written. I suggest stating whether there were group x time interactions for any of the variables. If there were significant group x time interactions, then state which groups differed over time and which did not. If there was no group x time interaction, then indicate whether there was a time main effect.
Table 1: Was the SpO2 of the divers actually that low (i.e., 58%)? Is it possible to have that level of desaturation? The typical SpO2 for breath-hold divers after a dive seems to be much higher (https://pubmed.ncbi.nlm.nih.gov/8286984/)
Line 253: “insignificant” – Please use “not significant” when referring to statistical analyses.
Line 254: It is unclear what “weakest” refers to here with regards to HR. Do you mean “lowest”?
Line 286: Could another reason for the greater decrease in HR in swimmers vs. runners be that the HR was much greater in the swimmers than the runner at baseline?
Add to the limitations section that females were not assessed in your study.
Comments on the Quality of English Language
The English needs minor revision
Author Response
Response to Reviewer 2 Comments
The authors would like to thank the reviewers for their interest in our work and for their insightful comments. We have carefully considered each of these recommendations and revised our manuscript accordingly using the “track changes” feature, and provided a point-by-point response to each of the individual recommendations below. We are appreciative of the reviewers’ time and energy.
Point 1: Line 16: Indicate what FVC stands for. I think it should be forced vital capacity?
Response 1: Thank you for your comment. The words vital lung capacity have been replaced by forced vital capacity.
Point 2: I am not sure the numbering is required in the abstract (i.e. (1), (2), (3), (4))
Response 2: The numbering in the abstract has been modified.
Point 3: Line 21: please indicate whether this was an increase or decrease in SPO2 (I assume decrease?)
Response 3: It has been added to the article.
Point 4: Line 22: Same comment for HR. Please indicate whether this is an increase or decrease.
Response 4: It has been added to the article.
Point 5: Lines 41-42: “The topicality of this topic...” This sounds awkward. I suggest re-wording.
Response 5: Thank you for your comment. The sentence has been rewritten.
Point 6: Lines 45-46: “Regular breathing exercises have a positive effect on increasing endurance and improving overall mental and physical health in athletes and the entire population.” References are needed to support this statement.
Response 6: Thank you for your comment. We have added a reference.
Point 7: Lines 48-51: I think references are also needed here to support the statements: “The quality of breathing plays a decisive role in performance in many sports disciplines. The direct effect on performance is particularly evident in the field of endurance sports, where the ability to maintain an optimal breathing rhythm can significantly improve results.”
Response 7: Thank you for your comment. We have added a reference.
Point 8: Line 79: “This insight provides valuable insights for investigating...” I suggest changing this to “This provides valuable insights for investigating...”
Response 8: Thank you for your comment. It has been added to the article.
Point 9: Lines 99-101: “The magnitude of the diving response, which is often measured using the bradycardia scale, appears to be inversely proportional to the decrease in arterial oxygen saturation” – should this be “proportional” and not “inversely proportional”?
Response 9: Thank you for your comment. The sentence has been rewritten.
Point 10: Lines 105-106: “However, endurance training is essential to achieve high performance in this sport.” – what sport are you referring to here?
Response 10: The sentence has been corrected.
Point 11: Lines 116-117: “...such training has been shown to increase blood hematocrit levels” – please provide a reference for this statement.
Please include a hypothesis statement (or statements) at the end of your introduction.
Response 11: Thank you for your comment. We have added a reference. We added a hypothesis at the end of the introduction.
Point 12: Line 166: As with the abstract, please spell out / explain the abbreviation FVC.
Response 12: Thank you for your comment. The words vital lung capacity have been replaced by forced vital capacity.
Point 13: Line 181: “FiO2 – 20.93%” – should this be “FiO2 = 20.93%”?
Response 13: Thank you for your comment. It has been corrected.
Point 14: When describing your p-values in the manuscript, make sure to use decimal places and not commas (i.e. p<0.05 and not p<0,05).
Response 14: Thank you for your comment. It has been corrected.
Point 15: The statistical analysis section is very unclear. This can be clarified by simply using a 2-factor (group x time, with repeated measures on the time factor) ANOVA with Bonferroni post-hoc testing if you have a group main effect or group x time interaction. It is not clear in the statistics section how the diving group was compared to the other groups. I suggest comparing the post-training results from the other groups to the diving group in a one-factor ANOVA. If this is significant, you can use a Bonferroni test to determine which groups are different.
Response 15: Thank you for your comment. The statistical analysis section was corrected. text The first sentence was a mistake in the text. We did the ANOVA analysis and the Bonferoni correction originally, but then we chose non-parametric analysis. We forgot to delete the sentence. Now we have corrected.
Point 16: Lines 211-212: “This change was significant (p<0,05) only for the swimmers and sedentary participants, not the runner group.” You should run a group x time ANOVA. You can only state that there were differences between groups over time if the group x time interaction is significant.
Response 16: Thank you for your comment. We performed the intergroup analysis using nonparametric methods. We used nonparametric methods because of number of participants.
Point 17: I think overall the results need to be more clearly written. I suggest stating whether there were group x time interactions for any of the variables. If there were significant group x time interactions, then state which groups differed over time and which did not. If there was no group x time interaction, then indicate whether there was a time main effect.
Response 17: Thank you for your comment. Since we had few participants in our research, we used non-parametric methods. Although these differences are non-significant, the graphical interpretation shows trends that we expect would be statistically significant if more participants were available.
Point 18: Table 1: Was the SpO2 of the divers actually that low (i.e., 58%)? Is it possible to have that level of desaturation? The typical SpO2 for breath-hold divers after a dive seems to be much higher (https://pubmed.ncbi.nlm.nih.gov/8286984/)
Response 18: Yes, it is possible. The divers were professionals. There are studies where divers reached even lower desaturation.
Point 19: Line 253: “insignificant” – Please use “not significant” when referring to statistical analyses.
Response 19: Thank you for your comment. It has been corrected.
Point 20: Line 254: It is unclear what “weakest” refers to here with regards to HR. Do you mean “lowest”?
Response: 20: Thank you for your comment. It has been corrected.
Point 21: Line 286: Could another reason for the greater decrease in HR in swimmers vs. runners be that the HR was much greater in the swimmers than the runner at baseline?
Response 21: The sentence was added to the text: Another possible explanation for why the swimmers' HR dropped so much is that after consulting their coach, three out of five swimmers had tachycardia.
Point 22: Add to the limitations section that females were not assessed in your study.
Response 22: Thank you for your comment. It was added to the article.
Reviewer 3 Report
Comments and Suggestions for Authors
This is an interesting study on how 8 weeks of apnea training may enhance variables related to performance and health. The study design is good, but there are several points in the manuscript that Authors need to revise.
1) Abstract provides a good overview of the study. Suggestion: Avoid using the same words from the title to the key-words.
2) Introduction: One of my main concerns here is the length of introduction and the lack of paragraphs. Intro is big and I cannot find the practical connection between increasing the ability of apnea and enhancing performance or health related variables. Perhaps, paragraphs will solve this problem. The introduction would be greatly benefit with paragraphs and research questions in an attempt to lead the readers to the main purpose.
One another important issue is the balance between the length of the intro and the length of the discussion. I suggest to Authors trying to balance these two.
3) Methods are good but my main concern here is that training was performed without supervision. Consequently, how the Authors are sure that athletes performed the training sessions? In line with this comment, was there any familiarization session for participants before entering the training period? Also, how Authors performed the supervision of the training sessions?
How many were the divers?
But, the basic question here is why the 30 sec were chosen as a time to perform apnea? Why 5 sets and the last MAX, and why 60sec of normal breathing between sets?
Figure needs a correction in the word apnea.
Line 205: Check the F value.
4) Results are ok. I suggest only adding the group abbreviation's explanations under the table 2.
5) Discussion similar to the intro will be greatly benefit from paragraphs. This will smooth the study and helps to a better comparison and discussion of the results.
I am not sure I read something regarding the positive effects on performance that the participants may have following the study. By the way, why Authors did not include a performance test inside the study?
In addition, this is a pilot study but there has to strong points and limitations. Therefore, Authors should add more limitations concerning the training program and perhaps more specific future directions.
The manuscript needs a good proof reading for grammatical errors as well as for the format of the paragraphs and the whole text (text alignment).
Author Response
Response to Reviewer 3 Comments
The authors would like to thank the reviewers for their interest in our work and for their insightful comments. We have carefully considered each of these recommendations and revised our manuscript accordingly using the “track changes” feature, and provided a point-by-point response to each of the individual recommendations below. We are appreciative of the reviewers’ time and energy.
Point 1: Abstract provides a good overview of the study. Suggestion: Avoid using the same words from the title to the key-words.
Response 1: Thank you for your comment. We changed the Keywords.
Point 2: Introduction: One of my main concerns here is the length of introduction and the lack of paragraphs. Intro is big and I cannot find the practical connection between increasing the ability of apnea and enhancing performance or health related variables. Perhaps, paragraphs will solve this problem. The introduction would be greatly benefit with paragraphs and research questions in an attempt to lead the readers to the main purpose.
Response 2: Thank you for your comment. We made paragraphs in the introduction and discussion.
Point 3: One another important issue is the balance between the length of the intro and the length of the discussion. I suggest to Authors trying to balance these two.
Response 3: Thank you for your comment. We expanded and slightly changed the discussion.
Point 4: Methods are good but my main concern here is that training was performed without supervision. Consequently, how the Authors are sure that athletes performed the training sessions? In line with this comment, was there any familiarization session for participants before entering the training period? Also, how Authors performed the supervision of the training sessions?
Response 4: Thank you for your comment. Participants sent information about their breath holding time each day immediately after the exercise. Once a week, the supervisor discussed and assessed each participants training and progress.
Point 5: How many were the divers?
Response 5: There were four professional divers. It was added to the text.
Point 6: But, the basic question here is why the 30 sec were chosen as a time to perform apnea? Why 5 sets and the last MAX, and why 60sec of normal breathing between sets?
Response 6: Thank you for your comment. We have added the source to the text and corrected it slightly.
Point 7: Figure needs a correction in the word apnea.
Response 7: The figure has been corrected.
Point 8: Line 205: Check the F value.
Response 8: Thank you for your comment. We corrected the number.
Point 9: Results are ok. I suggest only adding the group abbreviation's explanations under the table 2.
Response 9: Thank you for your comment. It was added to the text.
Point 10: Discussion similar to the intro will be greatly benefit from paragraphs. This will smooth the study and helps to a better comparison and discussion of the results.
Response 10: Thank you for your comment. We made paragraphs in the introduction and discussion.
Point 11: I am not sure I read something regarding the positive effects on performance that the participants may have following the study. By the way, why Authors did not include a performance test inside the study?
Response 11: Thank you for your comment. The measurement was time-consuming enough that we decided not to do a performance test in the pilot study, but we would like to add one in a wider study.
Point 12: In addition, this is a pilot study but there has to strong points and limitations. Therefore, Authors should add more limitations concerning the training program and perhaps more specific future directions.
Response 12: Additional limitations and future recommendations have been added to the article.
Round 2
Reviewer 1 Report
Comments and Suggestions for Authors
Dear Editor,
The present study has shown a substantial improvement in scientific content. However, the comparative statistics does not seem to be the most appropriate due to the sample size in each group. It is suggested that descriptive statistics be used and the discussion be speculative. Additionally, it is necessary to correct the text format.
Keywords:
It is suggested that the keywords (up to 5 words) be cited in alphabetical order.
Materials and Methods:
It is necessary for the authors to justify the adopted sample size, statistical methods, and scientific basis. How was the sample size per group determined? Inter-group statistical analyses with low sample numbers can result in Type I errors. Furthermore, was a normality test conducted? In this case, it is believed that it would be best to adopt descriptive statistics only.
Author Response
Response to Reviewer 1 Comments
Point 1: Keywords: It is suggested that the keywords (up to 5 words) be cited in alphabetical order.
Response 1: Thank you for your comment. We changed it, keywords are cited in alphabetical order.
Point 2: The present study has shown a substantial improvement in scientific content. However, the comparative statistics does not seem to be the most appropriate due to the sample size in each group. It is suggested that descriptive statistics be used and the discussion be speculative. Additionally, it is necessary to correct the text format.
Response 2: Thank you for your comment, it is true (the discussion has to be more spekulative). We corrected the discussion, it is more spekulative now.
Point 3: It is necessary for the authors to justify the adopted sample size, statistical methods, and scientific basis. How was the sample size per group determined? Inter-group statistical analyses with low sample numbers can result in Type I errors. Furthermore, was a normality test conducted? In this case, it is believed that it would be best to adopt descriptive statistics only.
Response 3: Thank you for your comment. We agree that the number of data in a group is not optimal, nevertheless the global envelope method that was used to analyse the data provide the graphical interpretation that is very useful for the reader to understand the results of the study, it can also be viewed as the graphically interpretted descriptive statistics (The solid line corresponds to the group means, the grey envelope determines the total variability). But the output also contains a few red points that identifies the differences between the groups on the common (global) significance level 0.05. The method provides the exact output (understand the type one error appears exactly in 5% of cases) for any kind of distribution and also for any number of data since it is based on permutations. See the cited paper please. Therefore we belive that the presented results are useful and correct.
The simple size is small, we know. Multiple measurements were impossible due to the nature of the experiment. We are aware of this, because of the robustness and strength of the method we are trying to get the maximum possible from a small sample.
Inter-group statistics was not count, it was our mistake. It was not included in the original version of the article, we misstated it based on the comments of the reviewer. We apologize and have removed it from the article. There is present statistical analysis for all participants together, in the table 1. We have specified the Methods section.
Normality of the data has been done, half of the data groups of parameters are not distributed normally. Thank you and a reminder, we have replaced the t-test with the Wilcoxon pair test. We corrected it in the method section.
Reviewer 2 Report
Comments and Suggestions for Authors
The authors have addressed my comments. I have a few minor suggestions for revision:
Lines 22-23: I suggest re-wording these lines to “SpO2 decreased by 6.8%, maximum apnea length increased by 15.8%, HR decreased by 9.1%, and FVC increased by 12.4% for the groups (p<0.05)”
Line 288: “The pre-intervention values of all intervention participants were significantly lower than those of professional divers” – I don’t think this statement is accurate for all of the variables.
Line 357: Change “gender” to “sex”
Comments on the Quality of English LanguageThe English needs minor revision
Author Response
Response to Reviewer 2 Comments
Point 1: Lines 22-23: I suggest re-wording these lines to “SpO2 decreased by 6.8%, maximum apnea length increased by 15.8%, HR decreased by 9.1%, and FVC increased by 12.4% for the groups (p<0.05)”
Response 1: Thank you for your comment. We corrected it.
Point 2: Line 288: “The pre-intervention values of all intervention participants were significantly lower than those of professional divers” – I don’t think this statement is accurate for all of the variables.
Response 2: Thank you for your comment. We corrected it. We meant only the FVC values, not all of them. It was added to the text.
Point 3: Line 357: Change “gender” to “sex”
Response 3: Thank you for your comment. We corrected it.
Reviewer 3 Report
Comments and Suggestions for Authors
-
Author Response
Thank you fro your review and your advice.